# Relationship between Vitamin D and Immunity in Older People with COVID-19

**DOI:** 10.3390/ijerph20085432

**Published:** 2023-04-07

**Authors:** Fulvio Lauretani, Marco Salvi, Irene Zucchini, Crescenzo Testa, Chiara Cattabiani, Arianna Arisi, Marcello Maggio

**Affiliations:** 1Department of Medicine and Surgery, University of Parma, 43121 Parma, Italy; 2Cognitive and Motor Center, Medicine and Geriatric-Rehabilitation Department of Parma, University-Hospital of Parma, 43126 Parma, Italy

**Keywords:** COVID-19, Vitamin D, U-shape, infections

## Abstract

Vitamin D is a group of lipophilic hormones with pleiotropic actions. It has been traditionally related to bone metabolism, although several studies in the last decade have suggested its role in sarcopenia, cardiovascular and neurological diseases, insulin-resistance and diabetes, malignancies, and autoimmune diseases and infections. In the pandemic era, by considering the response of the different branches of the immune system to SARS-CoV-2 infection, our aims are both to analyse, among the pleiotropic effects of vitamin D, how its strong multimodal modulatory effect on the immune system is able to affect the pathophysiology of COVID-19 disease and to emphasise a possible relationship between the well-known circannual fluctuations in blood levels of this hormone and the epidemiological trend of this infection, particularly in the elderly population. The biologically active form of vitamin D, or calcitriol, can influence both the innate and the adaptive arm of the immune response. Calcifediol levels have been found to be inversely correlated with upper respiratory tract infections in several studies, and this activity seems to be related to its role in the innate immunity. Cathelicidin is one of the main underlying mechanisms since this peptide increases the phagocytic and germicidal activity acting as chemoattractant for neutrophils and monocytes, and representing the first barrier in the respiratory epithelium to pathogenic invasion. Furthermore, vitamin D exerts a predominantly inhibitory action on the adaptive immune response, and it influences either cell-mediated or humoral immunity through suppression of B cells proliferation, immunoglobulins production or plasma cells differentiation. This role is played by promoting the shift from a type 1 to a type 2 immune response. In particular, the suppression of Th1 response is due to the inhibition of T cells proliferation, pro-inflammatory cytokines production (e.g., INF-γ, TNF-α, IL-2, IL-17) and macrophage activation. Finally, T cells also play a fundamental role in viral infectious diseases. CD4 T cells provide support to B cells antibodies production and coordinate the activity of the other immunological cells; moreover, CD8 T lymphocytes remove infected cells and reduce viral load. For all these reasons, calcifediol could have a protective role in the lung damage produced by COVID-19 by both modulating the sensitivity of tissue to angiotensin II and promoting overexpression of ACE-2. Promising results for the potential effectiveness of vitamin D supplementation in reducing the severity of COVID-19 disease was demonstrated in a pilot clinical trial of 76 hospitalised patients with SARS-CoV-2 infection where oral calcifediol administration reduced the need for ICU treatment. These interesting results need to be confirmed in larger studies with available information on vitamin D serum levels.

## 1. Introduction

Vitamin D is an umbrella term including a group of lipophilic hormones with pleiotropic actions. It has been traditionally related to bone metabolism, although several studies in the last decade have suggested its role on cardiovascular diseases, diabetes, malignancies, infections, autoimmune and rheumatic inflammatory diseases [1,2]. The two most important forms of vitamin D are known as vitamin D3 (cholecalciferol) derived by endogenous production, mainly driven by 7-dehydrocholesterol generation in the skin through UV-B solar irradiation or by dietary intake of animal products and vitamin D2 (ergocalciferol) derived from plants products.

To reach the biological active form, vitamin D3 and vitamin D2 need two hydroxylation reactions. The first one takes place in several tissues, mainly in the liver, converting them into 25(OH) vitamin D (calcifediol, 25(OH)D). The second one occurs in kidney, bone, immune cells and parathyroid glands, where 25(OH)D is converted into 1,25(OH)_2_ vitamin D (calcitriol) by CYP27B1, a mitochondrial enzyme. Calcitriol enters circulation and travels either to its major target tissues like bone, kidney and gastrointestinal tract or to unconventional ones, such as immune cells, parathyroid glands and epithelial cells, where it interacts with its nuclear receptors (vitamin D Receptors, VDRs).

Through these interactions, calcitriol exerts both its principal action on calcium/phosphate skeletal homeostasis and novel and its non-classic ones on muscle metabolism, cardiovascular diseases and immunity [3]. Notably, the effects on immunological system include regulation of the innate and adaptive immune responses [4]. Since vitamin D could have a potential role in infectious diseases, especially lung infections such as COVID-19 [5], our aims are:To analyse, among the pleiotropic effects of this hormone, how its strong multimodal modulatory effect on the immune system is able to affect the pathophysiology of this disease;To outline a possible relationship between its circannual fluctuations in blood levels and the epidemiological trend of COVID-19, particularly in the elderly population.

## 2. Epidemiology of Vitamin D Deficiency

Much debate has taken place over the definition of vitamin D deficiency [6]. 

The European Calcified Tissue Society Working Group has defined severe vitamin D deficiency as a calcifediol serum level lower than 30 nmol/L [7]. There is evidence that calcifediol concentration < 50 nmol/L, or 20 ng/mL, is an indicator of vitamin D deficiency, whereas a concentration of 51–74 nmol/L, or 21–29 ng/mL, indicate insufficiency; finally, serum levels >30 ng/L, or 75 nmol/L, suggest a status of vitamin D sufficiency [8]. This evidence is based on intestinal calcium absorption, which is maximised above 80 nmol/L, or 32 ng/mL, in postmenopausal women [9]. Independently of the cut-off value used to define vitamin D deficiency, low levels of calcifediol are extremely common, emerging as a global public health issue. The US National Center for Health Statistics estimates that about 70% of the population may have vitamin D3 deficiency [10], while another study showed that 13.0% of 55,844 European individuals had serum calcifediol concentrations < 30 nmol/L [11].

As we know, exposure to solar ultraviolet-B radiation is a major source of vitamin D3 production. In fact, vitamin D deficiency is particularly common during the winter and spring time (minimum levels in April and maximum ones in September), periods in which the vitamin D is stored in fatty tissues after exposure to the sunlight during the summer and autumn when this process has ended [12]. Isaia et al. reported circulating levels of calcifediol lower than 12 ng/L in 76% of Italian women over 70 during the winter [13]. Although vitamin D deficiency is more prevalent at higher latitudes, also elderly people living in Southern Europe experience a significant risk of vitamin D deficiency [9]. In fact, a recent population study performed in the Chianti area of the Italian region of Tuscany known for its temperate climate and sunny countryside, showed a high prevalence of vitamin D deficiency. Serum levels of vitamin D diminish with age in both sexes, but the decline is substantially earlier and steeper in women in the perimenopause period, while in men it occurs approximately 20 years later, starting in the seventh decade [14]. Older persons are particularly prone to developing hypovitaminosis D, and its estimated worldwide prevalence is about 50% [15]. This is due to specific physiological and lifestyle factors linked to advanced age, such as impaired production of 7-dehydrocholesterol in the skin, insufficient exposure to sunlight and UVB radiation (and/or excess clothing), poor dietary intake of vitamin D, as well as chronic diseases, pharmacological treatments and disability [16]. In a study conducted in a US cohort of older American men, both vitamin D deficiency and insufficiency were common. Approximately one fourth had calcifediol levels below the threshold of frank deficiency (<20 ng/mL), and the majority had vitamin D insufficiency (<30 ng/mL) [17].

Nevertheless, as in the general population, data on serum 25(OH)D concentration in the European elderly population are extremely heterogeneous, making it difficult to estimate the real prevalence of vitamin D deficiency. Different causes are responsible for the different quality of the data and include differences in analytical methods [18], with high variability even for the same assay technique [19]. The Vitamin D Standardization Program (VDSP) has developed protocols to standardise existing serum 25(OH)D data from different national surveys to address these methodological differences in the measurements of serum 25(OH)D concentrations. Application of the VDSP protocol for standardisation of serum 25(OH)D data from the two major surveys previously performed in European geriatric populations (LASA [20] and AGES-Reykjavik [21]) showed a prevalence of calcifediol deficiency (serum 25(OH)D < 30 nmol/L) of 4.6% and 8.4%, respectively [11]. The translation of these data into clinical practice is far to be applied because most of these studies have been conducted in Northern Europe, where fortification of foods with vitamin D is widely used.

In view of the worldwide prevalence of hypovitaminosis D, especially in the elderly, safe sun exposure should be favoured and adequate intake of vitamin D should be encouraged through the consumption of fortified foods as well as oral supplements.

## 3. Immunological Effects of Vitamin D

Among the pleiotropic effects of vitamin D, in the pandemic era, here we emphasise its strong role in modulation of the immune system, cellular proliferation and differentiation. The biologically active form of vitamin D, or calcitriol, can influence both the innate and the adaptive arm of the immune response.

### 3.1. Innate Response

The vitamin D level in the immune cells is regulated by two enzymes, whose activity depends on its concentration. The mitochondrial CYP27B1 enzyme can synthesize calcitriol, while the mitochondrial CYP24A1 enzyme is able to convert calcifediol and calcitriol into the inactive form of vitamin D. The calcitriol production is induced, in macrophages, by the stimulation of Toll-like receptors (TLRs) [22]. This activity is not regulated by parathormone (PTH) and growth factors, as FGF23, which acts directly on renal proximal tubules. Furthermore, in the macrophage cytoplasm exists an unfunctional form of CYP24, which does not allow the mitochondrial entry of vitamin D through seizing it and avoiding the subsequent inactivation from the active form of the enzyme [23]. The innate immune response activation against a pathogen starts from TLRs signalling in immune cells, such as polymorphonuclear leukocytes, dendritic cells, monocytes and macrophages through production of reactive oxygen species (ROS), cathelicidin and beta-defensin Cathelicidin is the most studied and its synthesis is induced by calcitriol through over-expression of the CYP27B1 enzyme, when cellular concentration of calcifediol is adequate [4]. Cathelicidin has several effects against both gram-positive and gram-negative bacteria, both enveloped and non-enveloped viruses and fungi [24], as well as immunomodulatory activity when is added to in vitro infected cell cultures. This antimicrobial power derives from the ability to disrupt bacterial membranes and to block the viral entry or replication by interfering with viral proteins [25].

In fact, TLRs activation in human macrophages increases the expression of the VDR and the 1α hydroxylase gene (CYP27B1), leading to the clearance of intracellular mycobacteria through cathelicidin production [22]. As proven by studies on murine models, the LL-37 peptide, the murine analogous of cathelicidin, reduces influenza A virus replication [26]. Other evidence suggests that cholecalciferol can modulate the production of inflammatory cytokines (TNF-α, INF-β, IL-8, IL-6) and chemokines (CXCL-8, CXCL-10), increasing the antiviral immune response [27]. In pneumocystis pneumonia, vitamin D supplementation can either reduce the production of TNF-α, INF-γ, IL-6 and inducible nitric oxide synthase (iNOS) or it can increase the expression of gene codifying antimicrobial peptides and proteins related to the anti-oxidation [28]. Both TNF-α and INF-γ can stimulate the production of calcitriol, suggesting an important role in macrophage host defence. The absence of vitamin D, VDR and CYP27B1 reduces the ability of the innate immune cells to produce antimicrobial peptides [4].

In conclusion, serum calcifediol levels, as suggested by the inverse correlation with upper respiratory tract infections, play a role in the innate immune response. Cathelicidin because of its chemoattractant action for neutrophils and monocytes [29], represents the first barrier in the respiratory epithelium to pathogenic invasion.

With ageing, the immune system can undergo progressive dysfunction and dysregulation, a phenomenon known as immunosenescence and inflammageing. Both these phenomena are characterised by a progressive reduction in the ability to drive effective immune responses against infections and by a low-grade systemic inflammation, respectively [30]. Furthermore, systemic inflammation in ageing is induced by different factors including immunosenescence, adiposity and gut inflammation. In this context, vitamin D could play a role, albeit with mechanisms not completely understood [31]. De Vita et al. reported a negative correlation between calcifediol and IL-6, supporting previous data reporting higher circulating levels of IL-6 in the VDR knockout mice, and highlighting a putative role for vitamin D in modulating inflammation, especially in the elderly [27]. A similar association was observed in community-dwelling older people by Laird et al., who found a significant reduction in plasma levels of CRP and IL-6 in subjects with sufficient calcifediol plasma concentration compared with those with insufficient levels. In the same study, there was a negative association between sufficiency status and the IL-6 to IL-10 ratio, indicative of a shift towards an anti-inflammatory Th2 immune response [32]. Another study conducted in an elderly population sample showed a negative correlation between circulating CRP and calcifediol serum levels; this association was stronger in patients with acute or chronic inflammatory diseases [33]. Finally, in a review of observational and intervention studies, the authors showed, an association between vitamin D and markers of inflammation, as well as an improvement in inflammatory status secondary to vitamin D supplementation [31].

### 3.2. Adaptive Response

Vitamin D exerts a predominantly inhibitory action on the adaptive immune response, and it influences either cell-mediated or humoral immunity through suppression of B cells proliferation, immunoglobulins production or plasma cells differentiation [34]. This role is played promoting the shift from a type 1 to a type 2 immune response [35]. The suppression of Th1 response is due to the inhibition of T cells proliferation, pro-inflammatory cytokines production (e.g., INF-γ, TNF-α, IL-2, IL-17) and macrophage activation [36]. Otherwise, vitamin D promotes the cytokine synthesis by Th2 lymphocytes, such as IL-4, IL-5 and IL-10; this phenomenon induces the indirect suppression of Th1 cells differentiation [37]. Moreover, Treg cells increase in number as well as in IL-10 concentration, which can block lymphocyte Th1 production [38]. In part, the effect of vitamin D on T cells proliferation derives from the interference with the mechanism of dendritic cells antigen presentation [39]. In fact, dendritic cells treated with vitamin D become tolerogenic and reduce their ability to stimulate T cells production both in vivo and in vitro [40]. The ability of vitamin D to modulate adaptive immune response has opposite implications for the treatment of autoimmune diseases (positive) [41], and infectious diseases (negative). T cells play a fundamental role in viral infectious diseases. CD4 T cells provide support to B cells antibody production and coordinate the activity of the other immunological cells; moreover, CD8 T lymphocytes remove infected cells and reduce viral load [42]. The importance of type 1 immune response, and the production of INF-γ and IL-17, was highlighted not only for the viral diseases, such as influenza infection, but also for bacterial pathology. In addition to the improvement of the Th2 response mediated by vitamin D, the enhancement of Treg lymphocytes and IL-10 production was associated with a worse prognosis in M. tuberculosis [43]. Currently, there is strong evidence that vitamin D supplementation could improve pulmonary infectious disease prognosis. However, it may have different effects, depending on either the branch of immune response or the type of micro-organism involved [25].

Such effects of vitamin D on adaptive immunity could be particularly useful in elderly individuals, in whom immunosenescence is characterised by a lack of naïve T cells, an abundance of differentiated memory T and B cells, alteration of physiological Th1/Th2 and Th17/Treg ratios, dendritic cell dysfunction, a reduction of peripheral B cell population, and their immunoglobulin or receptor repertoire. These abnormalities of the adaptive immune response may on the one hand promote a pro-inflammatory state and yet on the other hand predispose the elderly to the development of inflammatory and infectious diseases in relation to exposure to new antigens and pathogens [30,44].

## 4. Epidemiology of COVID-19 in Older Persons 

To date, more than 758 million individuals have been infected since the pandemic started, and almost 6.859 million people have died due to the virus [45].

The SARS-CoV-2 genomic sequence is 96% homologous to the bat coronaviruses genomes, but bat–human transmission has not yet been demonstrated. Genome sequence studies have generated the hypothesis that this virus has reached humans through an intermediate host, identified in pangolin, whose flesh was sold in China markets for consumption. Human–human diffusion subsequently spread the pandemic. The World Health Organization declared the dissemination of SARS-CoV-2 as a pandemic on 11 March 2020. Data from the WHO–China fact-finding mission showed that the overall case fatality rate (CFR) in February 2020 was 0.7% in contrast with individuals older than 80, whose population exhibited a CFR of 21.9% [46].

Table 1 shows the prevalence and fatality rate of COVID-19 in older persons.

Three different reports have considered the CFR in specific age and comorbidity groups. The first report showed an overall CFR of 2.3% in 72,314 patients, but its rise to 8% in individuals 70–79 years of age and 14.5% in those older than 80 years of age [47]. The second report analysed 4226 cases from the United States and reported a CFR < 1% in adults younger than 54 years of age, from 3 to 11% in patients 65–84 years of age and 10 to 27% in those older than 85 years of age. More than 80% of the deaths have occurred in individuals older than 65 years of age [48] and, similarly, the National Vital Statistic System (NVSS) of the US government reported 114,411 deaths for COVID-19 between May 1 and 31 August 2020 in 50 states plus the District of Columbia, and 78.2% of these were in patients older than 65 years [49]. In the third report, from China’s fact-finding mission, CFR was 1.4% in patients without comorbidities, 7.6% in those with cancer, 8% in those with chronic pulmonary disease, 8.4% in those with hypertension, 9.2% in those with diabetes and 13.2% in those with cardiovascular disease, highlighting the higher risk of severe disease and death for individuals older than 60 years of age and of those with comorbidities [50].

While the case fatality rate (CFR) is the deaths to reported cases ratio, the infection fatality rate (IFR) is the deaths to total cases of infection ratio and is intrinsically related to the age-specific infection model. The IFR rose from 1.4% at age 65 to 4.6% at age 75 and to 15% at age 85. COVID-19 deaths in New York were 1/10 of the reported cases but 1/100 of all SARS-CoV-2 infection cases [51].

Thus, the elderly population which is more exposed to COVID-19 is at higher risk of poor outcomes; in May 2020, 38.7% of infected patients were older than 70 years and 69.6% were older than 50 years in Italy [52]; 78.4% of deaths were in patients aged between 60 to 89, especially in nursing homes. In addition, in northern regions (1.6% of deaths in 1000 inhabitants in Lombardy), a greater incidence was observed than in southern regions [12].

Lower lymphocytes counts and serum albumin, higher lactate dehydrogenase, urea nitrogen and C-reactive protein blood levels, higher numbers of pulmonary lobes involved and a greater proportion of bacterial coinfection were reported in the elderly rather than in adults younger than 60 years. In addition, a poorer prognosis and a higher mortality were reported in older adults with SARS-CoV-2-induced myocardial damage than ones without the myocardial damage [53].

In a Chinese study, the incidence of malnutrition was higher in older adults with COVID-19 disease, reaching 50.5% for those admitted to rehabilitation institutions and 38.7% for those admitted to hospital [54]; this phenomenon contributes to the immunological dysregulation and may exacerbate the elderly susceptibility to this infectious disease [55].

## 5. Immunological Pathophysiology of COVID-19

The severity of COVID-19 is due to both the viral infection and the host response. The immune response fighting SARS-CoV-2 infection is sequentially activated in the upper airway tract and in the lung. As a consequence, the local recruitment of macrophages and lymphocytes, cytokines production is generated together with adaptive T and B cell response stimulation. All these processes aimed at the suppression of the pathogen in frail individuals with poor immune response, may cause either significant lung or systemic damage [56].

Little evidence is available regarding an elderly patient’s specific immune response to SARS-CoV-2 infection. Chronological and biological ageing are both characterised by altered function of both innate and adaptive immunity. These phenomena, known as immunosenescence together with inflammageing and individual immunobiography, could adversely affect the reactivity to the SARS-CoV-2 infection and increase the vulnerability to severe COVID-19 outcomes in older adults [57,58].

### 5.1. Innate Response

Innate immunity is the first line of defence against SARS-CoV-2 and its inefficacy is associated with a higher prevalence of viral infection and a poor prognosis [59].

The first interaction with SARS-CoV-2 is the recognition of single strand RNA by pattern recognition receptors (PRRs), cytosolic RIG-like receptors (RLRs) and extracellular or endosomal Toll-like receptors. This process activates the production of inflammatory cytokines such as TNF-α, IL-1, IL-6, IL-18 and interferons (INFs, type I/III), which are important for viral defence. These events stimulate host immune response against the virus infection and promote the subsequent adaptive response [60].

Coronaviruses can block the interferon signalling pathway and promote other inflammatory ways that generate cytokine storm and exacerbate the disease. SARS-CoV-2 is susceptible to interferon I/III treatment in vitro and the absence of a significant increase of interferons gene expression (INF signature) in SARS-CoV-2 infected cells, as was shown by D. Blanco-Melo et al., where the unbalanced gene expression of interferon I was present in patients with severe COVID-19 when compared with the mild or moderate cases of the pathology [61]. Age-related change in innate immune response may involve interferon I production, increasing the susceptibility to viral infection. Of note, Qian et al. found that the expression of interferon I genes was lower in dendritic cells of aged donors; moreover, the induction of interferon signalling proteins, such as STAT1, IRF7 and IRF1, was reduced, suggesting defective regulation of the protein [62].

The escape from immunological recognition and from the interferon pathway highlights the importance of the role of dysregulation of the interferon I response in the pathogenesis of COVID-19; time is one of the most important factors because interferon provides protection in the early phase of disease while it becomes pathological in the advanced stage [60]. The increased expression of ACE-2 induced by interferon in the airway’s epithelium could be an answer [63]. Other pathogen defensive mechanisms are evasion from PRRs recognition and antagonism of their function [64].

The myeloid cells response dysregulation is a key mechanism, which determines the typical characteristics of COVID-19, such as ARDS, cytokine storm and lymphopenia. Significantly, high plasmatic levels of pro-inflammatory cytokines like IL-6 were reported in various studies and show a relationship with disease severity [65]. Both power and duration of the myeloid cells signalling could influence the prognosis of COVID-19 [42]. Different studies reported the reduction of natural killer (NK) cell numbers in peripheral blood of individuals with SARS-CoV-2 infection, which is related to poorer outcomes [66].

Specific SARS-CoV-2 IgG antibodies can activate NK cells, through the recognition of the antibody’s FC component by specific cell receptors, linked either to a pathogen’s antigen expressed on the surface of infected cells or to circulating virions in the form of immunocomplexes. These interactions could induce both cytokine production from NK cells and infected cells lysis through antibodies dependent cellular cytotoxicity (ADCC) [67].

Mutual interaction between monocytes and NK cells could damage recognition and elimination of SARS-CoV-2 infected cells [42]. Innate immunity inefficacy is strongly associated with the SARS-CoV-2 primary infection loss of control and higher risk of fatal disease, accompanied by innate immunity cells pathology [68].

### 5.2. Adaptive Response

#### 5.2.1. Antibodies Production

Similar to MERS infection, SARS-CoV-2 disease severity is closely related to the activity of neutralising antibodies. This is especially true for secondary infection [69]. Passive immunisation with neutralising antibodies during infection has not produced a wide effect on disease outcomes, highlighting a crucial role played by T cells on the pathogen clearance [70]. The spike receptor binding domain (RBD) is the principal target of neutralising antibodies for SARS-CoV-2 [71]. Memory B cells selective for the spike receptor increase within 120 days from infection and then their number stabilises. Those selective for the RBD display a similar kinetic activity, with nucleocapsid specific memory B cells rising in the first 4–5 months from the symptom’s onset [72].

Immunoglobulin’s isotypes of spike specific memory B cells change over time during the infection. During the first phase (from twenty to sixty days from symptom onset), IgM and IgG representation is similar, while after 6 months from the symptom onset, IgM decline and IgG prevail. IgA positive memory cells represent approximately five percent of the entire memory cells population. Their number remains steady for eight months. Analogous behaviour (increasing IgG, short half-life IgM and stable IgA) is observed for the RBD and nucleocapsid specific memory B cells.

Circulating levels of either the spike protein or RBD (which is the most important target for neutralising antibodies) specific to IgG or the spike protein and RBD specific to memory B cells are higher in hospitalised patients, suggesting that both the long-term humoral immunity compartments against SARS-CoV-2 are stronger in individuals who experienced a more severe course of disease. Confining the pathogen in the upper airway tract could reduce the disease severity and represent a valid strategy for converting it in a common cold and/or making it asymptomatic [72]. A strong B cell response is stimulated by SARS-CoV-2, as is demonstrated through the wide and quick production of specific IgA, IgG and IgM during the days after infection [73]. Seroconversion usually occurs from 7 to 14 days after symptom onset and antibody titres persist in the weeks following the pathogen clearance. The RBD is highly immunogenic, and antibodies which links this antigen could neutralise and block the interactions between host receptors (ACE-2) and viral particles. Specific IgG for trimeric spike protein of SARS-CoV-2 is measurable in serum until 60 days after symptom onset, but its level starts to decrease from 8 weeks [74]. Memory B cells are formed during primary infection and can rapidly respond to reinfection through the production of new plasma cells which are highly affine. The induction of these two cell types gives long-term protection. In a patient with COVID-19, a seroconversion with strong antibody production and memory T cells response could offer protection to reinfection [42]. Several studies have shown that high virus specific antibody titres for SARS-CoV-2 are related to wider in vitro neutralisation and are inversely connected with the patient viral load. Despite these observations, higher antibodies levels are associated with more severe clinical cases, suggesting that a robust humoral response alone is not sufficient to avoid a serious disease [75].

These data create warnings concerning the possibility that the humoral response for this virus could contribute to lung pathology through antibody dependent amplification (ADE). This phenomenon consists of the facilitation of the virus access to cells, which express FC receptor, in particular monocytes and macrophages, leading to their inflammatory activation [76].

#### 5.2.2. Cell-Mediated Response

Patients with COVID-19 and convalescent patients show that a T cell response is associated with a reduction in disease severity, suggesting that a specific T cell response (CD4 and CD8) against SARS-CoV-2 could be important for the control and resolution of the primary infection [77]. Seventy percent of COVID-19 patients had CD8 T cells at one month from symptom onset, and this percentage decreased to fifty after six months; their half-life is about 125 days and the kinetics is comparable to the some other acute viral infections. Ninety-two percent of COVID-19 cases had circulating CD4 T cells at one month from symptom onset, while 92% had these cells after six months; CD4 T cells have a half-life of 94 days [78]. Follicular T helper (Th) lymphocytes are a specialised CD4 cell subtype, which is important for B cell activation, neutralising antibodies production and long-term humoral immunity promotion [79]. Specific Th lymphocytes for SARS-CoV-2 were associated with a reduction of COVID-19 disease severity and those specific for the spike protein were present in almost all cases at 6 months from symptom onset; a substantial Th follicular memory was also found at the same time point [80]. While memory CD8 lymphocyte titres do not show significant differences between hospitalised and non-hospitalised patients, memory CD4 cell levels tend to be low in hospitalised individuals. This difference in T cells behaviour suggests that hospitalised patients, who are individuals with a more severe disease course, could have a weaker T cell response in the acute phase of SARS-CoV-2 infection [77]. Lymphopenia was seen in moderate to severe cases of COVID-19 with a drastic reduction of CD4 and CD8 T cells count. Lymphopenia severity, more evident for CD8 T cells of patients admitted to intensive care units, is associated with higher severity and mortality of SARS-CoV-2 infection [81].

On the other hand, individuals with mild symptoms typically present a normal or slightly high T cells count [82]. Various mechanisms could contribute to the reduction of blood T cells level, including the effect of the environment generated by inflammatory cytokines. In particular, IL-6, IL-10 and TNF-α serum levels are positively correlated with lymphopenia. This relationship is supported by the normalisation of T cells count after the reduction of these cytokines in convalescent patients [83].

Another mechanism includes both the retention of T cells in lymphoid organs and the endothelial adhesion induced by cytokines as INF-γ and TNF-α, which inhibit the recirculation of lymphocytes from these sites to the bloodstream [84]. In an autoptic study analysing spleens and hilar lymph nodes of individuals who died from COVID-19, an intense lymphocytes cellular death was found, suggesting the potential etiologic role of IL-6, as well as FAS-FAS ligand interaction [85].

While the induction of robust cellular T immunity is probably essential for efficient virus control, the dysregulated cellular T response can contribute to disease severity in COVID-19 patients [86]. A loss of Treg could facilitate the development of pulmonary immunopathology in patients with SARS-CoV-2 [42] and reduce the number of T regulatory cells in severe cases of this pathology [87]. Cell function has been shown to be impaired in CD4 and CD8 T cells of critically ill patients, with reduced production of polyfunctional T cells, as well as cytokine synthesis. In particular, INF-γ and TNF-α are generally lower. In cases of severe COVID-19, the cytotoxicity of CD8 T cells is reduced, with a tendency towards exhaustion of all CD8 T cells as well as the loss of degranulation and production of granzyme B. In contrast, during recovering, patients show a rise in follicular CD4 T cells (Tfh) as well as a reduction of the titres of inhibitory markers together with an increase in circulating effector molecules such as granzyme A, B and perforin [66].

#### 5.2.3. Immunological Memory

The evaluation of immunological memory derived from SARS-CoV-2 infection requires the analysis of different components: B lymphocytes, antibodies production and CD8 and CD4 T cells. Their different and independent kinetics are important for understanding the protection duration against the primary infection [88].

In their study, Jennifer M. Dan et al. evaluated immunological memory in five compartments (IgA and IgG antibodies, memory B cells, CD4 and CD8 T cells) at 1–2 months after symptom onset of the primary infection, where the majority (64%) of the individuals had positive relief of all the five of these at 5–8 months. At the second point in the study, the number of individuals with positivity in the five compartments decreased to 43%, while the 95% had positive values for almost three of the compartments. The heterogeneity of the immune response could derive from low viral load or little initial inoculum of SARS-CoV-2, but the presence of positive relief of immunological memory branches in a majority of individuals, and for months following recovery, shows that long-lasting protection against COVID-19 is possible. However, the role of upper airway tract immunological memory after the primary infection and the susceptibility to early reinfection of individuals with frail immune response need to be more investigated [72].

## 6. Role of Vitamin D in COVID-19 

To date, several studies focused on the effectiveness of calcitriol to prevent the upper respiratory tract infections, which in particular are caused by the influenza virus [89,90]. Therefore, a protective role of vitamin D in preventing or modulating the SARS-CoV-2 infection has also been hypothesised. Furthermore, vitamin D deficiency has been associated with comorbidities including cardio-vascular disease, diabetes and upper respiratory tract infection. Providing evidence to this notion, adequate levels of vitamin D can reduce the risk of many chronic diseases that are well-known risk factors for severe SARS-CoV-2 disease [24]. There is increasing evidence that individuals with vitamin D deficiency have a higher relative risk of testing positive for SARS-CoV-2 virus compared with those with normal levels of vitamin D [91]. Table 2 shows the studies investigating the relationship between Vitamin D and COVID 19. Ilie et al. evaluated the average values of vitamin D deficiency in 20 different European countries. The authors found a negative correlation between vitamin D mean levels and number of cases of COVID-19/1 million people in each Country. In the same study, low vitamin D levels were associated with higher number of deaths caused by COVID-19/1 million people [89]. In a single-centre case-control survey in an unselected large cohort of patients with Parkinson’s disease, there was no significant difference in the risk and mortality related to COVID-19 disease from the general population. However, the patients taking vitamin D supplements were less likely to get COVID-19 [92]. D’Avolio et al. conducted a retrospective monocentric observational study on three cohort of patients in Switzerland (PCR-positive for SARS-CoV-2 patients, PCR-negative patients and an additional control cohort) with available data on calcifediol plasma concentrations. These authors found significant lower calcifediol levels (*p* = 0.004) in PCR-positive patients after stratifying patients by age >70 years [93]. In addition, other data indicate an association between reduced levels of vitamin D and clinical severity of COVID-19 disease [94]. In a retrospective observational trial, De Smet et al. investigated the association between serum levels of calcifediol at time of hospitalisation, radiological stage (from stage 1 to 3) and prognosis of pneumonia during SARS-CoV-2 infection. In particular, 59% of the ill patients (47% women and 67% men) presented vitamin D deficiency at the ward entrance. Moreover, the male patients showed progressively lower levels of calcifediol as the radiological stage progressed, with the degree of deficit growing from 55% of stage 1 to 74% of stage 3. The calcifediol deficiency at entry was not affected by age, ethnicity, chronic lung disease, coronary artery disease, hypertension or diabetes yet was associated with mortality. Pre-existing presence of chronic lung disease was positively associated with mortality while a strong association was observed between vitamin D deficiency and mortality (OR of 3.87), independent of coronary heart disease, chronic lung disease and diabetes [91]. These results suggest that vitamin D could play a protective role against the SARS-CoV-2 infection, deriving from its known anti-inflammatory and immunomodulatory properties. Moreover, Vitamin D deficiency, seems an independent major cardiovascular risk factor and a marker of pre-existing pneumopathy and heart disease. Vitamin D acts at different level, preventing the respiratory epithelium infections, stimulating the production of antimicrobial peptides and modulating the cytokine storm and the progressive lung injury, respiratory failure and acute respiratory distress syndrome (ARDS) [95].

According to these studies, vitamin D could reduce the risk of infection through different mechanisms:Strengthening the respiratory epithelial barrier gap junctions [96].Inducing the transcription of antimicrobial peptides such as cathelicidin and defensins that can reduce viral replication [25].Increasing antioxidant factors gene expression, in particular glutathione reductase, that preserves vitamin C levels, or modulating the Nrf2-Keap 1 pathway [97], on which vitamin A could also act [98], resulting in antioxidant and antimicrobial effects [37].Modulating the inflammatory response by both reduction of pro-inflammatory cytokines responsible for pulmonary epithelial damage and stimulating the production of anti-inflammatory cytokines [93].Modulating adaptive immunity towards a Th2 driven response [96].

There has been growing interest on the ability of vitamin D to act on the renin-angiotensin system. Specifically, calcitriol exerts an effect on the angiotensin converting enzyme 2 (ACE-2)/angiotensin (Ang)-(1-7)-Mas axis, resulting in ACE-2 overexpression [89].

ACE-2-Ang-(1-7)-Mas axis is juxtaposed with the ACE-Ang-II-AT_1_R axis, which is instead known for its pro-inflammatory effects. ACE-2 is able to induce anti-inflammatory and antifibrotic processes in the respiratory system, has anti-oxidative stress and protective effects on vascular function [99]. However, ACE-2 has been identified as the epithelial receptor for SARS-CoV-2, expressed in various human tissues. Its expression is higher in the lungs and gastrointestinal system, while is lower in vascular endothelial cells, heart and kidney [96]. SARS-CoV-2 can bind the host cell receptors through the spike protein (S protein), leading to fusion with the cell membrane and its internalisation [100]. Thus, it has been hypothesized that the vitamin D-induced ACE-2 overexpression could facilitate virus entry into the host cells. Previous studies have shown the association between high levels of ACE-2 and better outcomes in COVID-19 patients [89]. The dysregulation of RAAS may result in a massive production of pro-inflammatory Th1 cytokines and exacerbate pneumonia progression that can lead to fatal ARDS [89]. Several studies have demonstrated the link between the imbalance of RAAS and vitamin D system during COVID-19 infection. Low calcifediol serum levels have been associated with increased activity of RAAS and angiotensin II serum titres [24]. Likewise, vitamin D supplementation can mitigate the cytokine storm and the endothelial damage. Other studies have shown that vitamin D is able to reduce lung permeability in animal models of ARDS in which rats supplemented with vitamin D had milder ARDS symptoms and moderate lung damage compared to controls [95].

For this reason, calcifediol could have a protective role in lung damage, both modulating the sensitivity of tissue to angiotensin II and promoting overexpression of ACE-2 [101].

**Table 2 ijerph-20-05432-t002:** Correlations between vitamin D and COVID-19.

Authors	Immune System Response	Type of Study	Main Conclusion
**De Smet et al., 2020 [91]**	Stimulation of expression of cathelicidin and beta-defensin in respiratory epithelia. Protolerogenic and anti-inflammatory cytokine synthesis by inhibiting T helper 1 (Th1) proliferation and switching Th1 CD4 T cells and M1-polarised macrophages towards a type II immunity.	Retrospective observational study on 186 subjects with polymerase chain reaction (PCR)-confirmed SARS-CoV-2 infection hospitalised from 1 March 2020 to 7 April 2020 for COVID-19 pneumonia.	Individuals with vitamin D deficiency have higher relative risk of testing positive for SARS-CoV-2, compared with those with normal level of vitamin D.
**Fasano et al., 2020 [92]**	Statistical correlation.	Single-centre case-control survey.	Patients taking vitamin D supplements were less likely to develop COVID-19.
**Annweiler et al., 2020 [95]**	Prevention of respiratory epithelium infections. Enhanced production of antimicrobial peptides. Modulation of the cytokine storm involved in lung injury.	Quasi-experimental study conducted in one geriatric acute care unit dedicated to COVID-19 patients.	Vitamin D supplementation was associated with less severe COVID-19 and better survival in frail elderly. Vitamin D was able to reduce lung permeability in the animal models of ARDS in which rats supplemented with vitamin D had milder ARDS symptoms and moderate lung damage compared with controls.
**Ali et al., 2020 [96]**	Strengthening the respiratory epithelial barrier gap junctions and modulation of adaptive immunity towards a Th2 driven response.	Randomised trials and meta-analysis.	Vitamin D supplementation has been shown to have protective effects against respiratory tract infections.
**Grant et al., 2020 [37]**	Increasing antioxidant factors gene expression, in particular glutathione reductase, that preserve vitamin C levels, or modulating the Nrf2-Keap 1 antioxidant pathway.	Review.	Higher 25(OH)D concentrations reduce risk of infection and death from acute respiratory tract infections, including those from influenza, CoVs and pneumonia.
**Ilie et al., 2020 [89]**	ACE-2 overexpression.	Short communication, Review.	There is an association between high levels of ACE-2 and better outcomes in COVID-19.
**Cereda et al., 2021 [101]**	Protective role of calcifediol in lung damage, both modulating the sensitivity of tissue to angiotensin II and promoting overexpression of ACE-2.	Single-centre cohort study.	Very low 25(OH)D levels were highly prevalent and suggestive of deficiency among hospitalised patients with severe COVID-19.

## 7. Therapeutic Perspectives

It is not surprising that the 25(OH)D blood level gradually decreases after summer and drops to its minimum concentration during winter, exactly when the incidence of the airway’s infections is highest. Based on these premises, vitamin D supplementation has to be started early, before the cold season starts and when the sunlight exposure is decreasing, in order to reach protective titres against these infections.

The calcifediol concentration to be reached in order to reduce the risk of airway’s infection is between 20 and 30 ng/mL [102]. Vitamin D supplementation is generally safe, and the potential adverse reactions are rare. In fact, as stated by the American Institute of Medicine, there were no adverse effects with a daily supplementation of vitamin D < 10,000 IU/die [24]. Doses greater than 10,000 IU/die might be safe, even if that is much higher than necessary. In fact, only 1000–2000 IU may be needed to obtain an optimal effect on bone and immunity. Special attention should be given to patients with rare diseases such as tuberculosis and sarcoidosis, at high risk of hypercalcemia [103]. However, the higher the concentrations of 25(OH)D, the higher the protection; the optimal range seems to be 40–60 ng/mL. About 50% of the population should ingest at least 2000–5000 IU/die of vitamin D3 to reach this target [104]. Alternative sources in the literature reasonably suggest that taking 10,000 UI/die of cholecalciferol for one month is effective to achieve target circulating levels of 40–60 ng/mL of 25(OH)D; furthermore, after the first month, the dose can be reduced to 5000 IU/die as maintenance [37]. A meta-analysis of individual participant data of 10,933 participants collected from 25 randomised controlled trials has shown that vitamin D supplementation can protect against acute respiratory tract infections. Special advantage from Vitamin D supplementation, patients with baseline 25(OH)D levels <25 nmol/L) and those receiving daily or weekly dose, [102]. Promising results for the potential effectiveness of vitamin D supplementation in reducing the severity of COVID-19 disease was demonstrated by a pilot clinical trial of 76 hospitalised patients with SARS-CoV-2 infection where oral calcifediol administration reduced the need for ICU treatment. Larger studies with available data on baseline vitamin D levels are needed [105]. Annweiler et al. in their prospective study, after stratifying hospitalised patients for COVID-19 into three groups, the first receiving regular vitamin D supplementation during the previous 12 months, the second group, where Vitamin D was administered just after hospital admission and the last group not receiving any supplementation. No significant benefit over the untreated group was shown from patients who were supplemented after the diagnosis of infection, whereas regular vitamin D consumption before the hospitalisation was associated with a significantly reduced risk of 14-day mortality. Conversely, vitamin D supplementation after diagnosis of SARS-CoV-2 infection was not associated with an improvement in disease outcomes [95]. Results from a retrospective analysis conducted on 91 multimorbid patients showed a significant improvement in the composite outcome (need for transfer to intensive care unit and/or death from any cause) after two consecutive doses of 200,000 IU of cholecalciferol in patients affected by COVID-19 with three or more diseases [90].

A multicentric, double-blind, parallel-group, randomised, placebo-controlled study was conducted in Brazil on 240 patients with moderate-severe COVID-19 disease. The primary outcome was the length of hospital stay while secondary outcomes were intra-hospital mortality, admission to ICU and mechanical ventilation. The treated group received a single dose of 200,000 IU of cholecalciferol in 10 mL of peanut oil. A single dose of vitamin D3 did not significantly reduce the length of stay and had no significant impact on clinically relevant outcomes in patients hospitalised with moderate-severe disease despite the increased serum level of vitamin D after supplementation [106]. It should be considered that enzymes involved in the metabolism of vitamin D require magnesium as cofactor for their reactions, especially in the kidney and liver. Consequently, a combination treatment of vitamin D, magnesium and vitamin B12 could have a more protective effect against clinical deterioration in the SARS-CoV-2 infection. For this reason, 250–500 mg/die of magnesium is recommended in addition to vitamin D during a SARS-CoV-2 infection [107].

Older patients show different clinical characteristics than adults. They are more likely to be affected by multimorbidity, physical frailty or disability, malnutrition (particularly of trace elements) and polypharmacy. Additionally, they also experience reduced sun exposure and low dietary intake of vitamin D and other micronutrients because of their lower ability to absorb and metabolise them, especially in the presence of medications negatively affecting kidney and liver function. Adequate supplementation of vitamin D and additional trace elements capable of strengthening the immune response is maximally important, especially considering the population at higher risk of developing severe COVID-19 and experiencing adverse outcomes.

## 8. Novel Vitamin D3 Hydroxyderivatives

In vitro and in vivo studies suggest new pathways of vitamin D3 metabolism to define the intermediates products and their potential biological activities. In addition to the most well-known biologically active form of vitamin D, calcifediol, of considerable interest are the metabolites of vitamin D derived from enzymatic cleavage by CYP11A1. CYP11A1-mediated metabolism of vitamin D involves sequential hydroxylation starting predominantly at C20 or C22 [108]. By these pathways, different hydroxyl derivatives of D3 are produced, such as 20(OH)D3 or 22(OH)D3; they could have anti-proliferative and anti-inflammatory effects, as well as be able to act on both the VDR and retinoic orphan acid receptors (ROR) α and γ [109]. Furthermore, the major product of secosteroidal pathways (20(OH)D3) is safe at high doses in mice without signs of toxicity and does not cause hypercalcemia, unlike conventional metabolites of vitamin D [108].

We underline that because of their systemic (adrenal gland) and local synthesis (immune system) [110], independent of hepatic metabolism, measurement of CYP11A1-derived vitamin D products in human serum could be additional and helpful to guide treatment. This information allows to go beyond a simple and single measurement of 25(OH)D, used in the past as the exclusive proxy of hormonal status (sufficiency or deficiency). For the same reason, transdermal, inhalatory or parenteral administration of vitamin D could result in the direct supply of synthesis precursors at the mentioned sources of CYP11A1-hydroxyderivatives. The immune system, in particular, represents an even more powerful stimulus to respond against inflammatory diseases such as the SARS-CoV-2 infection.

Their potential therapeutic role in therapy is even more interesting when we consider the impaired hepatic metabolism of vitamin D occurring in elderly individuals. Since these molecules are non-toxic and non-calcemic at relatively high doses [111], they represent a promising treatment, especially in this population, compared with the canonical oral administration of vitamin D.

## 9. Conclusions

Recommendations for nutritional supplementation of immunomodulatory elements, such as vitamin D, should be strengthened. 25(OH)D levels should be hopefully assessed before starting supplementation. Only individuals who need extra vitamin D should be supplemented, avoiding unnecessary costs and following the guidelines available on bone health (32 ng/mL is the recognised threshold of 25(OH)D needed to avoid secondary hyperparathyroidism) [103]. We should also remind that the only well-codified therapeutic target to be reached is that related to bone health and osteoporosis. Reaching the above-mentioned therapeutic target brings vitamin D concentration considerably closer to the serum hormone level capable of giving an immunity boost. In Southern Europe, in Countries with similar climates or in those where foods aren’t fortified with vitamin D, the supplementation should be started before winter, from September in particular (a month in which the blood level is lower), to reach a protective serum concentration in the cold season and reduce the risk of severe COVID-19 disease (Figure 1) [24]. Vitamin D supplementation could be stopped in April (a month in which solar UVB irradiation allows endogenous production of vitamin D) until adequate blood levels are reached. Individuals with a frail immune response or who are unable to reach a sufficient solar exposure, such as frail elderly people, should receive continuous vitamin D supplementation to ensure a persistent optimal serum level. No mega-doses are needed because they could lead to an increased risk for adverse effects, while small daily doses are sufficient to have a boost effect on immunity.

In the general population, and particularly in the elderly one, and based on vitamin D circannual fluctuations, we addressed the main predisposing factors for its rampant deficiency and its multimodal modulatory effect on the immune system. In particular, we focused on the SARS-CoV-2 host-pathogen interaction, COVID-19 epidemiological and pathophysiological elements.

As shown in Figure 1, especially in the elderly population, the relationship between vitamin D deficiency and SARS-CoV-2 infection is U-shaped. We also outlined the potential role of this pleiotropic hormone, during the pandemic era, in modifying the natural history of this disease and in the armamentarium of therapeutic effective strategies to counteract COVID-19.

## Figures and Tables

**Figure 1 ijerph-20-05432-f001:**
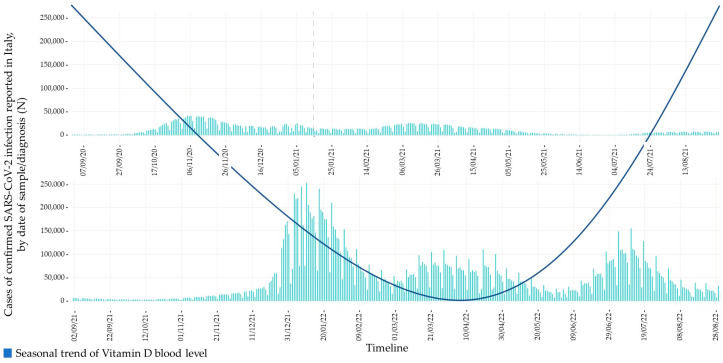
Hypothesised U-shaped relationship between protective serum concentration of vitamin D in the cold season and the risk of the SARS-CoV-2 infection. Modified from: The COVID-19 Task force of the Department of Infectious Diseases and the IT Service, Istituto Superiore di Sanità. COVID-19 integrated surveillance data in Italy. Available online: https://www.epicentro.iss.it/en/coronavirus/sars-cov-2-dashboard (accessed on 1 March 2023).

**Table 1 ijerph-20-05432-t001:** Epidemiology of COVID-19 in older persons.

Authors	Epidemiological Data	Conclusion
**Shahid et al., 2020 [46]**	China’s case fatality rate (CFR) in February 2020 was 0.7%, in contrast with individuals older than 80, whose population exhibited a CFR of 21.9%.	The SARS-CoV-2 pandemic has a much higher mortality rate in older adults.
**Wu et al., 2020 [47]**	In a report of 72,314 cases from the Chinese Center for Disease Control and Prevention, the overall case-fatality rate (CFR) was 2.3%, but it rose to 8% in individuals 70–79 years of age and 14.5% if older than 80 years of age.	CFR was elevated among those with pre-existing comorbid conditions—10.5% for cardiovascular disease, 7.3% for diabetes, 6.3% for chronic respiratory disease, 6.0% for hypertension and 5.6% for cancer.
**Bialek et al., 2020 [48]**	In a report of 4226 cases from the United States, the clinical fatality rate (CFR) was 1% in adults younger than 54 years of age, from 3 to 11% in patients 65–84 years of age and 10 to 27% in those older than 85 years of age.	More than 80% of the deaths occurred in individuals older than 65 years of age.
**Gold et al., 2020 [49]**	In a report of 114,411 persons who died from COVID-19 in the United States, the percentage of decedents aged ≥65 years was 77.6%.	Persons aged ≥65 years are disproportionately represented among COVID-19–associated deaths.
**WHO [50]**	In a report of the WHO-China Joint Mission on Coronavirus Disease 2019, the clinical fatality rate was 1.4% in patients without comorbidities, 7.6% in those with cancer, 8% with chronic pulmonary disease, 8.4% with hypertension, 9.2% with diabetes and 13.2% with cardiovascular disease.	Individuals older than 60 years old have a higher risk of severe disease and death.
**Levin et al., 2020 [51]**	The infection fatality rate (IFR) in a meta-analysis of 27 studies rose from 1.4% at age 65, to 4.6% at age 75 and 15% at age 85.	COVID-19 is hazardous not only for the elderly but also for middle-aged adults.
**ISS [52]**	In an Italian report, 38.7% of infected patients were older than 70 years and 69.6% were older than 50 years; 78.4% of deaths were in patients aged between 60 to 89.	The elderly population is more exposed to COVID-19 and at higher risk of poor outcomes.

## Data Availability

Not applicable.

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
