# Peer review of "Relationship between Vitamin D and Immunity in Older People with COVID-19"

_ijerph, 2023, doi:10.3390/ijerph20085432_

Round 1

Reviewer 1 Report

The manuscript is very well written and its content is very interesting.

I ask the authors to enrich some of the correlations set out in the individual paragraphs in a summary table to make the understanding of the correlations between vitamin D and covid 19 more immediate.

Author Response

Dear Reviewer,

We now revised estensively point by point the manuscript according to your suggestions.

We hope that the paper could be suitable for publication on IJERPH.

Best Regards,

Fulvio Lauretani,

University of Parma

Reviewer 2 Report

The manuscript ijerph 216976 is a well written review on the role of vitamin D in the course of Covid-19 disease and its involvement in immunity in general. Only minor points should be revised for clarification.

1)    page 8, line 395: please delete “whose” from the sentence

2)    page 8, line 400: please correct in PCR-positive patients

3)    page 9, lines 424: could vitamin D not only affect vitamin C but also other antioxidants or antioxidant pathways such as vitamin A?

4)    page 9, line 430: please rephrase “Th2 tolerogenic response”, maybe Th2 driven response would fit better as Th2 immunity is not considered as tolerogenic overall

5)    page 9, line 442: please rephrase “vitamin D-induced overexpression of ACE-2”

6)    page 10, line 505: additionally, calcium is essential for calcitriol synthesis

Author Response

(The authors gave the same response as above.)

Reviewer 3 Report

This study concerns the relationship between Vitamin D and immunity in older people with COVID-19. However, the descriptions provided by the authors were not well described as below.

1. Epidemiology of vitamin D deficiency and immunological effect of vitamin D. These two sections should focus more on the deficiency and immunological effect of vitamin D in older people compared to the facts of vitamin D deficiency and its immunological effect on the general population.

2. The authors should state if there were any differences in the immunological pathophysiology of COVID-19 in older people.

3. The authors should add more studies that relate the Vitamin D and older people. In the text. There is only one study that mentioned Vitamin D linked to older age (Line 396 to 401). This paper lacks the review of the relationship between Vitamin D and immunity in older people.

4. “The Calcifediol deficiency at entry was not affected by age, ethnicity, chronic lung disease, coronary artery disease, hypertension, diabetes and was associated with mortality” (Line 408-410) was contradicted by “These results suggest that Vitamin D could play a protective role against SARS-CoV-2 infection, deriving from its known anti-inflammatory and immunomodulatory properties (Line 413-415). Please explain.

5. Therapeutic perspective also should focus more on older people and explain the difference between younger and older people if available.

6. Line 470-471: “However, the higher the concentrations of 25(OH)D, the higher the protection and the optimal range seems to be 40 – 60 ng /mL contradicted with the conclusion (Line 532-534) “the guidelines available on bone health (32 ng/mL is the recognized threshold of 25(OH)D needed to avoid secondary hyperparathyroidism)”. Please explain.

6. The conclusion did not mention the relationship between Vitamin D and immunity in older people with COVID-19.

Author Response

(The authors gave the same response as above.)

Reviewer 4 Report

1) Abstract. Among the pleiotropic effects of Vitamin D, in the pandemic era, here we emphasise its ever-stronger role in modulation of immune system, cellular proliferation and differentiation. The biologically active form of Vitamin D, or Calcitriol, can influence both the innate and the adaptive arm of the immune response. Please, Underline the the aim of the study

2) Abstract.  Promising results for the potential effectiveness of Vitamin D supplementation in reducing the severity of COVID-19 disease was demonstrated in a pilot clinical trial of 76 inpatients with SARS-CoV-2 infection where oral Calcifediol administration reduced the need for ICU treatment. These interesting results need to be confirmed in larger studies with available information on Vitamin D serum levels. Please, correct the typo "inpatients" and underline the novelty of the study.

3) Introduction. L 38-40. Vitamin D is an umbrella term including a group of lipophilic hormones with plei-  otropic actions. It has been traditionally related to bone metabolism, although several  studies in the last decade have suggested its role on cardiovascular diseases, diabetes,  malignancies, autoimmune diseases and infections. Please ameliorate this paragraph and add several references, such as:

a- Vitamin D deficiency and clinical correlations in systemic sclerosis patients: A retrospective analysis for possible future developments. PLoS One. 2017;12(6):e0179062.  doi:10.1371/journal.pone.0179062

b- Trabecular Bone Score and Bone Quality in Systemic Lupus Erythematosus Patients. Front Med (Lausanne). 2020;7:574842. Published 2020 Sep 30. doi:10.3389/fmed.2020.574842

4) Introduction. L56-57. Notably, the effects on immunological system include regulation of the innate and adaptive immune responses [1]. Thus, Vitamin D could have  a potential role on infectious diseases, especially lung infections, such as COVID-19.

a- The Interaction of Vitamin D and Corticosteroids: A Mortality Analysis of 26,508 Veterans Who Tested Positive for SARS-CoV-2. Int. J. Environ. Res. Public Health 202219, 447. https://doi.org/10.3390/ijerph19010447

b- c- Shift Work and Serum Vitamin D Levels: A Systematic Review and Meta-Analysis. Int. J. Environ. Res. Public Health 202219, 8919. https://doi.org/10.3390/ijerph19158919

5) Introduction L 57. Thus, Vitamin D could have  a potential role on infectious diseases, especially lung infections, such as COVID-19. Please, add a brief description of study aim. 

6) 4. EPIDEMIOLOGY OF COVID-19 IN OLDER PERSONS. Please, add a table to present the most important articles and to summarise their conclusions.

7) L 523-526. Measurement of non-calcemic CYP11A-derived Vitamin D in human serum could  aid, despite a single measurement of 25(OH)D, in determining Vitamin D status of suffi-  ciency or deficiency. Since some of these molecules are non-toxic and non-calcemic at rel- atively high doses, they could also be potential therapies for treatment of inflammatory  diseases such as SARS-CoV-2 infection. Please, clarify this paragraph.

8) Conclusions. L538-543. It could be stopped in April, month from which solar UVB  irradiation allows endogenous production of Vitamin D, until adequate blood levels are  reached. Individuals with a frail immune response or unable to reach a sufficient solar  exposure, such as frail elderly people, should receive continuous Vitamin D supplemen-  tation, to ensure a persistent optimal serum level. No mega-doses are needed; this could  lead to an increased risk for adverse effects. Small daily doses are enough to have a boost  effect on immunity. Please, improve the description of conclusions and underline the novelty of this of this paper.

8) Please, add a figure/flowchart to explain the  vitamin D mechanisms in covid 19. 

Author Response

(The authors gave the same response as above.)
